# Z-Scheme Heterojunction of Phosphorus-Doped Carbon Nitride/Titanium Dioxide: Photocatalytic Performance

**DOI:** 10.3390/molecules29184342

**Published:** 2024-09-12

**Authors:** Jinyu Yang, Yanglin Zhang, Kun Liu, Dongxu Tang, Shizhong Zhou, Xiaojie Yang, Yuesheng Li, Yi Liu

**Affiliations:** 1Key Laboratory of Coal Conversion and New Carbon Materials of Hubei Province, School of Chemistry and Chemical Engineering, Wuhan University of Science and Technology, Wuhan 430081, China; 18727791310@163.com; 2School of Nuclear Technology and Chemistry & Biology/Hubei Key Laboratory of Radiation Chemistry and Functional Materials, Hubei University of Science and Technology, Xianning 437100, China; zyl619433573@163.com (Y.Z.); 17754119632@163.com (K.L.); tdtd800@126.com (D.T.); 15820177382@163.com (S.Z.); mailyangxiaojie@126.com (X.Y.); 3School of Chemical and Environmental Engineering, Wuhan Polytechnic University, Wuhan 430023, China

**Keywords:** Z-type heterojunction, photocatalytic degradation, photosensitive antibacterial

## Abstract

With increasingly serious environmental pollution problems, the development of efficient photocatalytic materials has become a hotspot in current research. This study focused on phosphorus-doped carbon nitride/titanium dioxide (PCT) Z-type heterojunctions, aiming to deeply investigate their photocatalytic degradation and photosensitive antimicrobial properties. A PCT Z-type heterojunction was successfully fabricated using melamine phosphate, cyanuric acid, and titanium dioxide. The structure, morphology, and optical properties of PCT Z-type heterojunctions were explored by FTIR, XRD, XPS, BET, SEM, UV-Vis DRS, TEM, EIS, and PL. A comprehensive and in-depth analysis of the structure, morphology, and optical properties of PCT Z-type heterojunctions was carried out. The photocatalytic degradation experiments revealed that PC3T Z-type heterojunctions exhibited an excellent degradation capability for methylene blue (MB) under visible light. The effect of PC3T on the adsorption–photocatalytic degradation of MB is more than 1.5 times that of a single titanium dioxide and P-doped carbon nitride. In the photosensitive antimicrobial performance study, PC3T reduced the survival rate of *E. coli* to 7%, after 120 min. Through free radical trapping experiments, it was shown that the hydroxyl radicals and superoxide radicals exerted an influence on the photocatalytic process. This study offers new ideas and approaches to address environmental pollution problems and holds significant theoretical and applied value.

## 1. Introduction

As industrialization continues to advance and the scale of urbanization expands rapidly, environmental pollution is becoming increasingly severe, and water pollution has become a key issue that needs to be solved urgently. Traditional methods for environmental treatment have many deficiencies in addressing such problems [1,2]. They are not only inefficient but also consume a large amount of energy. More seriously, new issues like secondary pollution will arise from the treatment process, which fails to meet the high-level environmental protection demands of modern society [3,4,5]. As a green and efficacious emerging technology, photocatalysis has substantial advantages in environmental governance [6]. In the process of pollutant degradation, the reaction conditions of photocatalysis are relatively mild and do not require high pressure, which can realize effective energy saving [7]. In terms of antibacterial research, photocatalytic technology can effectively eliminate pathogenic bacteria in the environment and will not cause secondary pollution [8,9]. Hence, the application prospect of photocatalysis technology in the environmental field is extremely wide, especially in resolving the problem of dye pollution and bacterial transmission.

Among numerous photocatalysts, titanium dioxide (TiO_2_) has become the research hotspot because of its distinctive properties and excellent performance [10,11]. TiO_2_ has excellent chemical stability, thermal stability, and photocatalytic activity [12]. Upon the irradiation of TiO_2_ by a photon with energy exceeding its bandgap width, electrons were stimulated to move from the valence band to the conduction band. This led to the formation of electron (e^−^) and hole (h^+^) pairs and subsequently resulted in the generation of reactive oxygen species (ROS) [13]. ROS has a powerful redox capacity and is capable of decomposing organic pollutants into harmless carbon dioxide and water, thus attaining the photocatalytic degradation of pollutants [14]. However, TiO_2_ has a large forbidden bandwidth and can only absorb ultraviolet light leading to its low utilization of solar energy. What is more, e⁻ and h^+^ on TiO_2_ are easily compounded, reducing the photocatalytic efficiency [15]. In order to break through these limitations, researchers are constantly exploring new modification approaches, among which building a heterojunction is an efficient approach.

Carbon nitride (g-C_3_N_4_), as a novel type of non-metallic photocatalyst, has become a hot research subject due to its distinctive structure and properties [16,17]. g-C_3_N_4_ has a suitable energy band structure and is capable of absorbing visible light, which endows it with an advantage in the utilization of solar energy [18,19]. Additionally, g-C_3_N_4_ has excellent thermal and chemical stability, and its preparation method is relatively simple and low in cost [20]. It has been mentioned to be the case that the combination of g-C_3_N_4_/TiO_2_ heterojunctions not only achieves a remarkable broadening of the light absorption range but also effectively facilitates the separation and transfer of photogenerated e^−^ and h^+^, which results in the enhancement of photocatalytic performance [21]. The band structure of g-C_3_N_4_/TiO_2_ belonged to a type II heterostructure [22,23]. However, some studies suggested that it might be a Z-type heterostructure [24]. Compared with the type II heterostructure, the Z-type heterostructure showed more significant superiority. In the Z-type heterostructure, the transfer paths of photogenerated e^−^ and h^+^ were more unique, endowing it with stronger redox ability. Ning et al. demonstrated that the creation of a Z-type heterojunction between g-C_3_N_4_ and TiO_2_ was capable of generating hydroxyl and superoxide radicals, while a type II heterojunction could produce only one type of radical [25]. The doping of elemental phosphorus offered new possibilities for the photocatalytic activity of the g-C_3_N_4_/TiO_2_ heterojunction. Based on previous reports, the main raw materials, methods, and applications for the preparation of phosphorus-doped carbon nitride/titanium dioxide composite catalysts are summarized and listed in Table 1. Su et al. prepared the heterogeneous structures of phosphate-doped carbon nitride/titanium dioxide nanotube arrays (P-CN/TiO_2_ NTs) by electrochemical anodizing, wet dipping, and hot polymerization [26]. P-CN/TiO_2_ NTs possessed preferable light absorption properties, boosted charge separation and transfer capability, and exhibited good photocatalytic performance for methylene blue (MB). Wadhai et al. prepared phosphorus-doped carbon nitride/titanium dioxide composites (PCN-P25) by the calcination and coupling method using melamine, 1-hydroxyethane-1,1-diphophonic acid, ethylene glycol, and P25 as raw materials [27]. PCN-P25 could be applied for hydrogen production. In addition, Kumar et al. prepared composite catalysts for hydrogen production by the hydrothermal method using various feedstocks [28]; Cako et al. prepared composite photocatalysts for CO_2_ reduction by the solid sublimation and conversion method [29]. However, their preparation processes were complex or used a wide variety of raw materials. Therefore, it is imperative to formulate a straightforward and practicable approach for preparation.

In this study, a phosphorus-doped carbon nitride/TiO_2_ (PCT) Z-type heterostructure was synthesized through an in situ method with melamine phosphate, cyanuric acid, and titanium dioxide as raw materials. The objective was to combine the respective advantages of phosphorus doping with carbon nitride and TiO_2_ to achieve more efficient photocatalytic degradation and antimicrobial effects. FTIR, XRD, XPS, BET, SEM, UV-Vis DRS, TEM, EIS, and PL were employed to conduct a comprehensive and meticulous analysis of its structure, morphology, and optical properties. Through photocatalytic degradation, photosensitive antimicrobial, and free radical capture experiments, the photocatalytic degradation and antimicrobial performance as well as the related mechanisms were investigated thoroughly, with the aim of providing new ideas and effective methods for addressing the environmental pollution problems.

## 2. Results and Discussion

The bonding characteristics were explained through FTIR spectra as shown in Figure 1b. The absorption peak at 607 cm^−1^ was attributed to the stretching oscillation of the Ti-O bond [30]. Moreover, the absorption peak at 3400 cm^−1^ was designated to the stretching band of the -OH group, indicating substantial adsorption of water molecules on TiO_2_ [31]. The FTIR spectrum of the PC disclosed distinctive stretching modes of the C-N heterocycle system in the range of 1200–1700 cm^−1^, as well as the heptazine units at 810 cm^−1^ [32]. The multiple bands within the range of 3000–3500 cm⁻^1^ were ascribed to N-H and -OH [33]. In PC3T, the featured peak of the CN heterocycle was observed. Although no heptazine units were detected, the absorption peak near the Ti-O bond broadened, suggesting that the composite of PC and TiO_2_ was successful.

Raman spectroscopy also demonstrated the successful preparation of PC3T, as shown in Figure 1c. The Raman spectra of TiO_2_ displayed five characteristic Raman bands (2*E*_g_, 2*B*_1g_, and *A*_1g_), among which the strongest *E*_g_ band was detected at approximately 144 cm^−1^ [27]. All the characteristic peaks belonging to TiO_2_ were maintained in PC3T. In the Raman spectra of TiO_2_, there was no hump seen around 1500 cm⁻^1^. After coupling with PC, the broad peak (978, 1228, 1345, and 1560 cm^−1^) between 900 cm^−1^ and 2000 cm^−1^ and all the peaks of TiO_2_ (140–800 cm^−1^) were observed, implying the formation of PC3T (interior illustration) [34,35]. However, the Raman peaks of PC3T exhibited slight shifts to lower wavenumbers in comparison with TiO_2_, attributed to the interaction of PC with TiO_2_. This might further confirm the variation in the electronic energy level of TiO_2_ after being combined with PC.

Figure 1d presents the XRD patterns of TiO_2_, PC, and PC3T samples illustratively. The crystal structures of TiO_2_ were mixtures of anatase and rutile crystals. The characteristic peaks were in correspondence with the standard data of JCPDS: 021-1272 and JCPDS: 021-1276 [36]. The significant peak at 27.23° corresponded to the (002) plane and the interlayer stacking structure of the aromatic compound [37]. Additionally, the peak at 13.01° was in connection with the (100) plane and was due to the in-plane repeat period of tri-s-triazine [38]. Due to the low PC doping, no distinct characteristic peaks were observed in PC3T. However, the characteristic peak width at 27.30° was broader than that of TiO_2_. This might be attributed to the presence of strong peaks for PC.

N_2_ adsorption–desorption isotherms were measured to conduct an analysis of the surface areas and porosity. As shown in Figure 1e, the specific surface areas of TiO_2_, PC, PC1T, PC2T, PC3T, and PC4T were 35.43 m^2^/g, 9.13 m^2^/g, 12.22 m^2^/g, 16.43 m^2^/g, 21.34 m^2^/g, and 20.15 m^2^/g. Compared to PC, PC3T had a larger specific surface area, which might be due to the incorporation of TiO_2_. In addition, PC3T had a large pore width relative to the other samples (Appendix A). The large specific surface area and pore width of PC3T facilitated the exposure of more active sites for the adsorption–degradation of dyes and photosensitized antibacterial activity [39,40].

As observed in Figure 2a, TiO_2_ showed evenly dispersed tiny spherical particles in aggregates, with a uniform diameter [41]. The PC exhibited irregularly stacked morphologies (Figure 2b) [42]. Figure 2c revealed that regular spherical nanoparticles were embedded on the surface of PC. The large number of TiO_2_ was uniformly dispersed and well-embedded in PC. At the same time, the element mapping indicated that Ti, O, C, N, and P were uniformly present on the surface of PC3T (Figure 2d).

The surface constituents and chemical statuses of the samples were examined through XPS. The survey scan of XPS presented the typical spectra of TiO_2_, and no other peaks were observed except for Ti, O, and C (Appendix A). PC possessed the peaks of C, N, O, and P. PC3T included all the characteristic peaks of the aforementioned TiO_2_ and PC. To acquire deeper insights into the molecular architecture and interatomic chemical bonding in the composite, the high-resolution spectra of Ti 2p, O 1s, N 1s, and P 2p of the samples are shown in Figure 3a–d. The bi-peaks of Ti 2p_3/2_ and Ti 2p_1/2_ belonging to PC3T were present in the Ti 2p peaks (Figure 3a). The energy interval between the double peaks was 5.66 eV, which was a characteristic of a typical Ti^4+^ species with a Ti-O structure [43]. Moreover, in contrast to TiO_2_, the binding energies of Ti 2p shifted negatively from 458.99 and 464.64 eV to 458.63 and 464.32 eV, respectively. In the O 1s spectrum of CP3T, three peaks at 529.80, 531.87, and 533.33 eV, correspond to the lattice oxygen, oxygen vacancy (O_v_) of TiO_2_, and the surface adsorbed oxygen. Likewise, the binding energy of adsorbed oxygen in PC3T had a positive shift compared to that in TiO_2_, which might be caused by the addition of PC. However, the lattice oxygen and the O_v_ in TiO_2_ had a negative shift from 532.04 and 533.35 eV to 531.87 and 533.33 eV, respectively. Binding energy shifts in XPS spectra indicated a strong interaction between TiO_2_ and PC in PC3T [44]. As depicted in Figure 3c, regarding the XPS spectrum of N 1s, three peaks at 398.61 eV, 398.81 eV, and 399.91 eV were, respectively, ascribed to the C-N-C, N-(C)_3_, and -NH_X_ bonds [45]. Figure 3d presents the high-resolution XPS of P 2p for PC3T and the peaks at 133.5 eV. As reported, the binding energies of P 2p in P-N and P-C bonds were approximately 133.5 eV and fall within the range of 131.5–132.5 eV, respectively. Therefore, P might tend to enter the g-C_3_N_4_ network as P-N in place of the C site, which might promote the redshift of the visible absorption [46].

TEM images further revealed the successful preparation of PC3T (Figure 3e–h). As illustrated in Figure 3e,f, the particle had a lattice spacing of 0.35 nm, aligning with the (101) crystal planes of anatase TiO_2_ [47]. After the formation of the composite photocatalysts, the morphology and lattice spacing of TiO_2_ remained largely unchanged. However, disordered structural features appeared around TiO_2_, echoing the characteristics of PC [48]. Therefore, PC3T had been successfully prepared.

There existed a strong connection between the performance and the photo-absorption of the photocatalysts. As depicted in Figure 4a, the UV-vis DRS absorption spectra of TiO_2_ merely absorbed within the UV irradiation range of about 410 nm. PC had a steep visible absorption peak around 450 nm [49]. The absorption peak was shifted toward visible wavelengths by the in situ binding of the two materials. Moreover, PC3T had a tail peak with an absorption of about 500 nm. This tail absorption peak could be an intermediate energy level formed between the conduction and valence bands, allowing electrons to be excited more easily and improving photocatalytic performance [23,50]. The bandgaps of the samples were identified using the Kubelka–Munk function method and are shown in Figure 4b. The bandgaps of TiO_2_, PC, and PC3T were 2.97 eV, 2.58 eV, and 2.84 eV.

The valence band (VB) positions of TiO_2_ and PC were determined by valence band XPS spectra (Figure 4c,d). The VB XPS of TiO_2_ indicated the VB position at 2.79 eV, whereas the VB of PC was at 1.80 eV. Based on the following formula, *E*_VB, NHE_ = *ϕ* + *E*_VB, XPS_ − 4.44, the work function (*ϕ*) of the instrument was 4.42 eV. Thus, the actual values of the valence band positions of TiO_2_ and PC were 2.77 V and 1.78 V, respectively. The determination of the conduction bands (CBs) was achieved by extrapolating Mott–Schottky curves. Figure 4e,f displays the Schottky curves of TiO_2_ and PC. Through extrapolating the curve to 1/*C*^2^ = 0, the Fermi level (E*_F_*) values of TiO_2_ and PC were determined as −0.30 and −0.90 V, respectively (vs. Ag/AgCl) [51,52]. These values were further changed to −0.10 and −0.7 eV (vs. NHE). It was found that the conduction band potential (E_CB_) of an n-type semiconductor was about 0.1 eV lower than the value of E_F_. The E_CB_ potentials of TiO_2_ and PC were, respectively, determined to be −0.20 V and −0.80 V. The bandgap was determined with the following formula: *E*_CB_ = *E*_VB_ + *E*_g_. The findings that integrated the analysis of bandgaps and valence band positions were in accordance with the conduction position [53]. Figure 4g displays the band structures of both TiO_2_ and PC.

The photogenerated charge transfer was intimately connected to the interface resistance, and this connection could be manifested through electrochemical impedance spectroscopy (EIS) [54]. The Nyquist plots of TiO_2_ and PC are presented in Figure 4h. The EIS results disclosed that PC3T had a smaller impedance arc radius compared to TiO_2_ and PC, signifying the lower charge transfer resistance.

Furthermore, the PL spectra of TiO_2_ and PC showed a clear emission peak at around 300 nm (Figure 4i). This was due to the recombination of photogenerated electrons and holes. The PL intensity of TiO_2_ was lower than that of PC. The PL spectrum of PC3T appeared to be shifted to the right, and the intensity was lower than that of TiO_2_ and PC. This phenomenon successfully demonstrated that the combination of TiO_2_ and PC promoted electrons and holes, preventing the recombination of photogenerated carriers, and thereby improving photocatalytic performance [55].

The photocatalytic degradation performance of MB catalyzed by various catalysts is shown in Figure 5a. The respective photocatalytic degradation rates of TiO_2_, PC, PC1T, PC2T, PC3T, and PC4T were 54%, 36%, 56%, 68%, 82%, and 59%. PC3T demonstrated exceptional adsorption–photocatalytic degradation performance. This could be attributed to the combined effect of a larger specific surface area and the creation of heterogeneous structures. The cycling test was conducted to assess the recycling potential of PC3T (Figure 5b). The result indicated that no significant change occurred in its degradation capacity after three cycles, confirming the outstanding recyclability of PC3T [56]. The photosensitive antibacterial properties of the materials were studied using *E. coli* as model bacteria. After 120 min, it could be seen from Figure 5c that PC3T showed outstanding antibacterial performance under illumination conditions, and the survival rate of *E. coli* was reduced to 7%. The related practical images can be found in Figure 5d.

As illustrated in Figure 5e,f, no signals of hydroxyl (•OH) and superoxide (•O_2_^−^) radicals could be observed under dark conditions. Nevertheless, the appearance of the characteristic peaks of DMPO-•O_2_^−^ and DMPO-•OH under light irradiation indicated the production of •OH and •O_2_^−^ in the reaction system, which was consistent with the characterization of the Z-type heterojunction for the production of radicals. What is more, the signals of •O_2_^−^ were weaker compared to those of •OH, suggesting that •OH radicals probably had a greater effect on the photocatalytic process than •O_2_^−^ radicals. Overall, •OH radicals significantly participated in the photocatalytic process of PC3T.

Based on the aforementioned results, Figure 6 presents the diagram of the photocatalytic degradation and antimicrobial mechanism of PC3T. XPS, TEM, etc., demonstrated the existence of better bonding between PC and TiO_2_ to form a heterogeneous structure. From the perspective of an energy band structure, when PC was combined with TiO_2_, a Z-type heterojunction was created at their interfaces. Under visible light excitation, both TiO_2_ and PC generate electrons (e⁻) from the valence band (VB) to the conduction band (CB) and retain holes (h⁺) in the VB. When the two photocatalysts are in close contact, part of the e⁻ of TiO_2_ can be rapidly transferred to the VB of PC, combining with the h⁺ of PC, and then excited into the CB of PC. The CB of the PC is positioned above the level at which reactive oxygen species are produced, so the e⁻ on the CB of the PC can interact with the O_2_ adsorbed on the surface to form ·O_2_^−^. In addition, the holes in VBs of TiO_2_ can interact with water or hydroxide ions (OH⁻) to form ·OH [57,58]. Consequently, the produced free radicals could achieve good degradation of MB and antimicrobial effects.

## 3. Conclusions

The above results suggested that it was feasible and effective to prepare PCT Z-type heterojunctions using melamine phosphate, cyanuric acid, and titanium dioxide through an in situ method. The characterization-based in-depth analysis of the structure, morphology, and optical properties of the composite photocatalysts laid the foundation for comprehending their structure and performance. BET tests demonstrated that PC3T had a large specific surface area and pore size. UV-Vis DRS evidenced a significant redshift of PC3T toward visible light. PL and EIS synergistically proved that photogenerated electrons and holes were well separated in PC3T. The photocatalytic degradation experiments confirmed the excellent degradation ability of PC3T Z-type heterojunctions for MB under visible light and the better photosensitive antibacterial activity. In terms of the adsorption–photocatalytic degradation of MB, the effect of PC3T is over 1.5 times greater than that of a single TiO_2_ and P-doped carbon nitride. Regarding the photosensitive antimicrobial performance, after 120 min, PC3T was able to reduce the survival rate of *E. coli* to 7%. ·O_2_^−^ and ·OH were proved to be the active species in the photocatalytic process by the EPR test, which was also in accordance with the characteristics of the Z-type heterojunction for the generation of free radicals. PCT Z-type heterojunctions were promising as effective materials for solving environmental pollution problems.

## 4. Experimental Sections

### 4.1. Materials

Melamine phosphate and melamine were acquired from Shanghai Macklin Biochemical Technology Co. (Shanghai, China). Cyanic acid was purchased from Shanghai Aladdin Reagent Co. (Evonik, Germany). TiO_2_ was obtained from Degussa GmbH in Germany. Beef extract, sodium chloride, peptone, agar powder, and yeast powder were purchased from Sinopharm Chemical Reagent Co. (Shanghai, China). Anhydrous sodium sulfate was provided by Aladdin Chemical Reagents Ltd. (Shanghai, China). Lyophilized powders of *E. coli* were acquired from the Henan Provincial Engineering and Technology Research Centre for Industrial Microbial Strains in China. Ultrapure water was used in the experiment.

### 4.2. Sample Preparation

As shown in Figure 1a, the composite photocatalysts were fabricated by an in situ method. Firstly, a certain mass of melamine phosphate was dissolved and then the cyanuric acid solution was added and stirred for 30 min. Secondly, 0.1 g of TiO_2_ was added and stirred for 3 min, and then centrifuged and dried. Subsequently, the solution was filtered and dried. Eventually, the powders were calcined at 500 °C for 1 h with the aim of obtaining PC1T, PC2T, PC3T, and PC4T, respectively. PC1T, PC2T, PC3T, and PC4T corresponded to different masses of cyanuric acid (0.129 g, 0.193 g, 0.258 g, and 0.323 g), respectively.

### 4.3. Material Characterization

The functional groups were ascertained by means of a Nicolet IS10 FTIR spectrometer (Gangdong Technology Development Co., Ltd., Tianjin, China) (in the range of 4000–400 cm^−1^). The crystal structures were identified through X-ray powder diffraction with the DMAX-D8X instrument (Shimadzu Co., Ltd., Kyoto, Japan). The morphologies were characterized via FE-SEM (Carl Zeiss AG Co., Ltd., Oberkochen, Germany) and FE-TEM (Thermo Fisher Scientific, Waltham, MA, USA). The molecular structures of samples were utilized for characterization through X-ray photoelectron spectroscopy (Thermo Fisher Scientific Co., Ltd., Waltham, MA, USA). The specific surface area and pore size distribution of the materials were gauged by a specific surface area analyzer (Quantachrome Instrument Co., Ltd., Boynton Beach, FL, USA). Raman spectroscopy was employed to verify the structural alterations in the samples using the Hrobioa Xplra plus Raman system (HORIBA Instrument Co., Ltd., Oberursel, Germany). UV–visible diffuse reflectance spectroscopy was measured within the range of 200–2000 nm by means of UH4150 (Hitachi Instrument Co., Ltd., Hitachi, Japan). The photoluminescence spectra of the materials were measured by the FLS1000 fluorescence spectrometer (Edinburgh Instruments Co., Ltd., Edinburgh, UK). Electrochemical measurements were performed by means of an electrochemical workstation equipped with a three-electrode setup (Shanghai Chenhua Instrument Co., LTD., Shanghai, China). Pt and Ag/AgCl electrodes were, respectively, employed as the counter and reference electrodes. A 0.5 M sodium sulfate solution was used as an electrolyte to test the samples for electrochemical impedance and Mott–Schottky measurements. The Bruker A300 could be used to obtain electron spin resonance (ESR) signals (Bruker Instrument Co., Ltd., Billerica, MA, USA).

### 4.4. Photocatalytic Test

The 10 mg samples were placed in 50 mL of the 10 mg L^−1^ (2.67 × 10^−5^ mol L^−1^) MB solution and shaken in the darkness for 40 min to achieve the adsorption–desorption equilibrium. Next, the solution was exposed to radiation from a 500 W Xenon lamp that had a filter (λ = 420 nm), and 4 mL of the suspension was withdrawn every 20 min. The supernatant was obtained by centrifugation. The absorbance of the MB solution at 664 nm was measured by a UV-Vis spectrophotometer (Shanghai Youke Instrument Co., Ltd., Shanghai, China). The degree of MB degradation was determined by absorbance values.

Cyclic stability experiments were conducted to demonstrate the reusability of the sample. The experimental procedure was similar to the photocatalytic degradation experiment described above. After the first photocatalytic degradation experiment, the photocatalyst was centrifuged, washed, and dried, and then the second photo–photocatalytic degradation experiment was carried out. After the same steps, the third photocatalytic experiment was carried out.

### 4.5. Photosensitive Antibacterial Experiment

The samples (5 mg mL^−1^) and the *E. coli* suspension (C ≈ 1.2 × 10^5^ CFU mL^−1^) were exposed to irradiation beneath a 200 W incandescent lamp. Every 60 min, the bacteria were aspirated and diluted for coating. The bacteria were cultivated for 15 h and then counted.

## Figures and Tables

**Figure 1 molecules-29-04342-f001:**
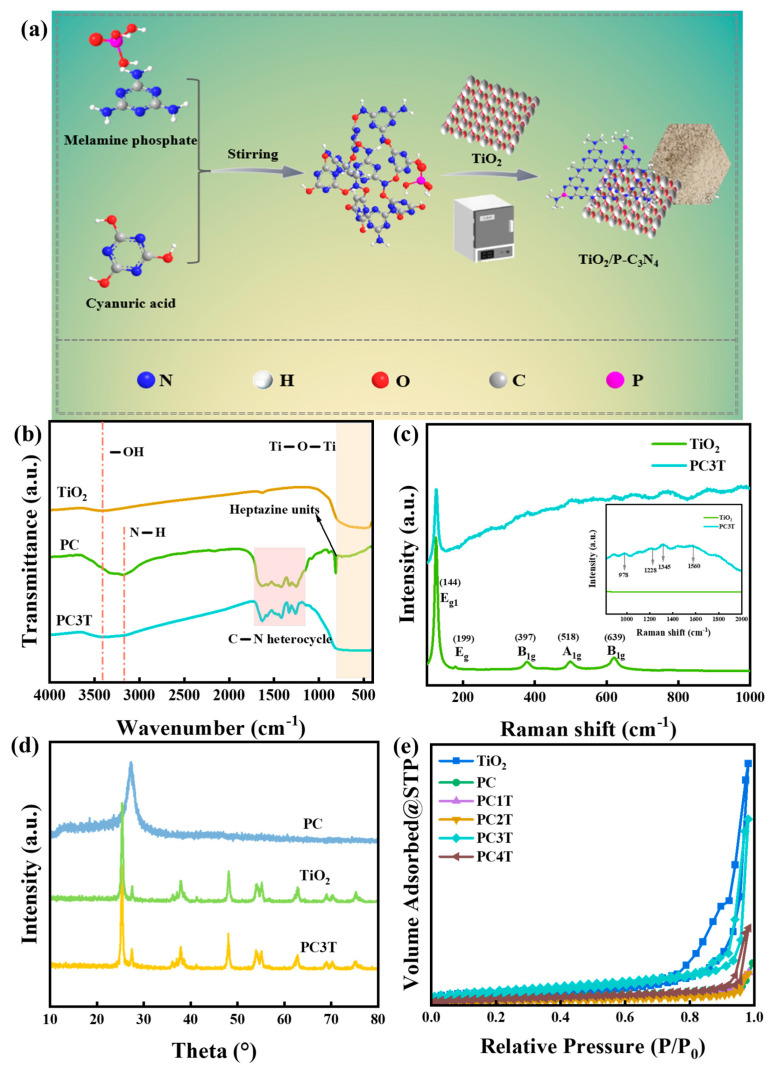
(**a**) Schematic illustration of PC3T, (**b**) FTIR spectra, (**c**) Raman spectra, (**d**) XRD patterns, and (**e**) N_2_ adsorption–desorption isotherms of various samples.

**Figure 2 molecules-29-04342-f002:**
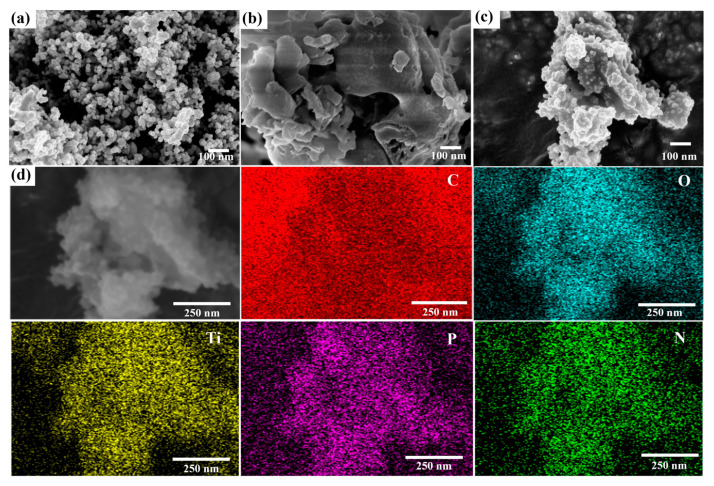
SEM images of (**a**) TiO_2_, (**b**) PC, and (**c**) PC3T; (**d**) HAADF-STEM image and corresponding EDS elemental (C, O, Ti, P, and N) mapping of PC3T.

**Figure 3 molecules-29-04342-f003:**
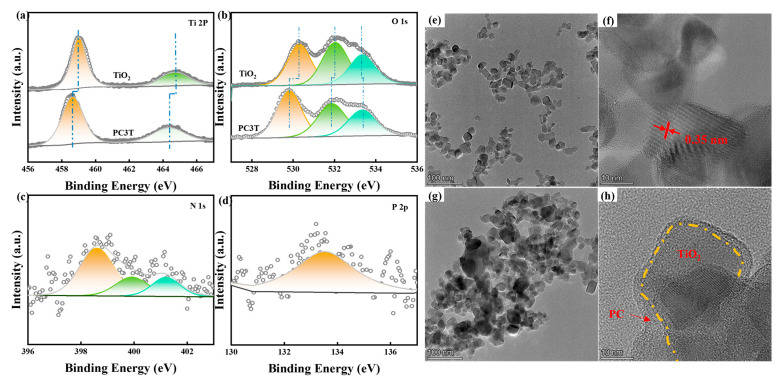
(**a**) Ti 2p and (**b**) O 1s spectra of TiO_2_ and PC3T; (**c**) N 1s and (**d**) P 2p spectra of PC3T; and TEM images of (**e**,**f**) TiO_2_ and (**g**,**h**) PC3T.

**Figure 4 molecules-29-04342-f004:**
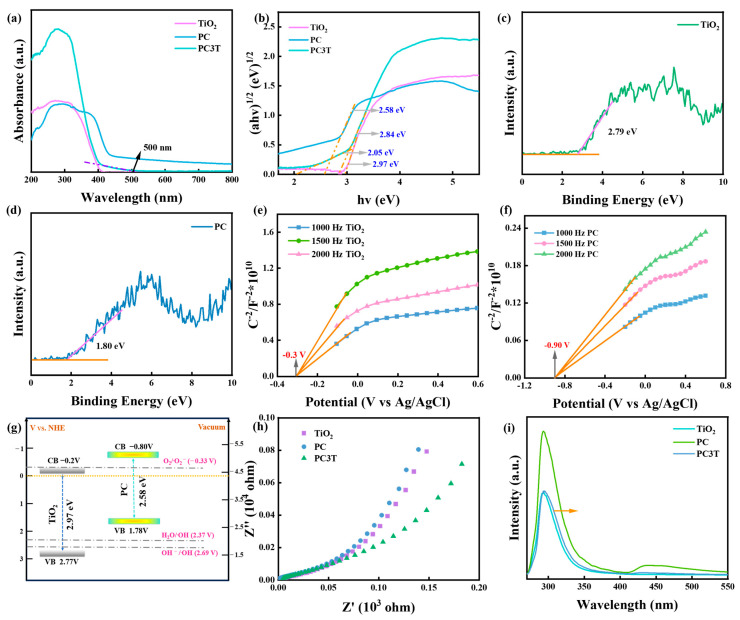
(**a**) UV-Vis DRS and (**b**) bandgap energy spectra of the samples; (**c**,**d**) valence band spectra of TiO_2_ and PC3T; Mott–Schottky plots of (**e**) TiO_2_ and (**f**) PC3T; (**g**) band structures of TiO_2_ and PC; and (**h**) EIS plots and (**i**) PL spectra of samples.

**Figure 5 molecules-29-04342-f005:**
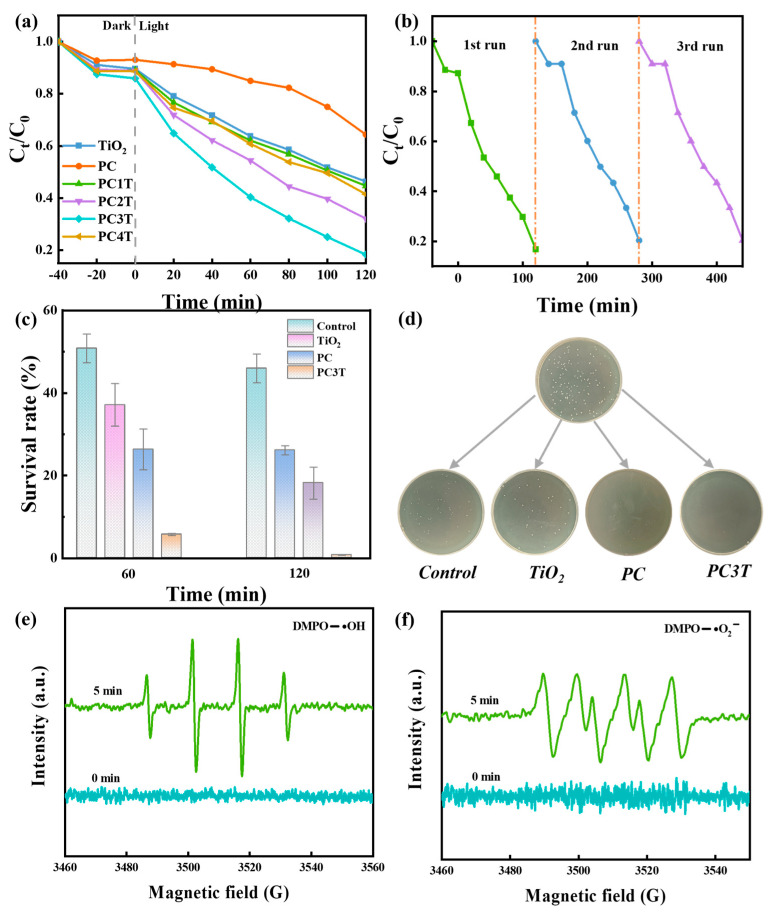
(**a**) Photocatalytic degradation of MB for various samples; (**b**) stability of PC3T for the photocatalytic degradation of MB; (**c**) photosensitive antimicrobial survival and (**d**) real images of control, TiO_2_, PC, and PC3T against *E. coli*; and ESR spectra of (**e**) DMPO-•O_2_^−^ and (**f**) DMPO-•OH for PC3T.

**Figure 6 molecules-29-04342-f006:**
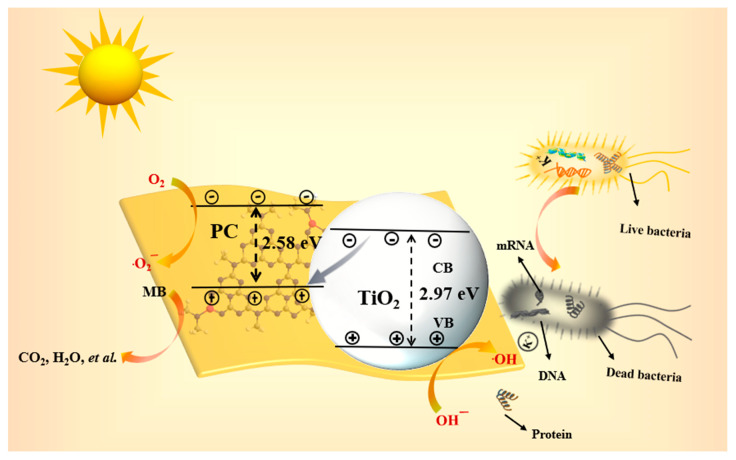
The diagram of the possible mechanism of photocatalytic degradation and photosensitive antibacterial for PC3T.

**Table 1 molecules-29-04342-t001:** Summary of materials, methods, and applications of phosphorus-doped carbon nitride/titanium dioxide composites.

Materials	Preparation Methods	Application Fields	References
Ammonium fluoride, dicyandiamide, 2-aminobenzonitrile, 1-butyl-3-methylimidazolium hexafluorophosphate, ethylene glycol	Anodizing and wet-soaking method	Degradation and hydrogen production	[26]
Melamine, 1-hydroxyethane- 1,1-diphophonic acid, ethylene glycol	Calcination and coupling method	Hydrogen production	[27]
Urea, citric acid, 1-butyl-3 methylimidazolium hexafluorophosphate, hydrofluoric acid, acetic acid	Hydrothermal method	Hydrogen production	[28]
Ammonium fluoride, glycerol, phosphate, urea, sodium sulfate	Solid sublimation and conversion method	CO_2_ reduction	[29]

## Data Availability

Data are contained within the article and Appendix A.

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
