# Peer review of "Z-Scheme Heterojunction of Phosphorus-Doped Carbon Nitride/Titanium Dioxide: Photocatalytic Performance"

_molecules, 2024, doi:10.3390/molecules29184342_

Round 1

Reviewer 1 Report

Comments and Suggestions for Authors

This manuscript “Z-Scheme Heterojunction of Phosphorus-Doped Carbon Nitride/Titanium Dioxide: Photocatalytic Performance » by Yang et al. describes the design of an efficient photocatalytic material, made of P-doped carbon nitride/titanium dioxide (PCT) as a Z-type heterojunction, for water decontamination by degradation of dyes (Methylene Blue) and microbes (E. Coli). PCT Z-type heterojunction was successfully fabricated using melamine phosphate, cyanuric acid, and titanium dioxide through an in-situ method.

The topic of water remediation is of major importance and this work has some interesting points (the in situ preparation method for example). The experimental work is done correctly. Except on some aspects, the manuscript is suitably written (see corrections after). I recommend the work for publication in the journal after minor corrections.

Hereafter, the main points are listed for necessary improvements before publication.

1) Something is missing in the introduction and in the discussion of obtained results. Please add how the proposed system compares itself/benchmarks to other systems in the field? For example, the information could be displayed by a few lines in the text, supported by a table listing the types of systems and their data (Z-heterojunction, preparation method, performances, references).

2) Something is missing in the conclusion. Please add what currently limits the reported system and your perspectives for improvement. This is important for an impactful report in the related field.

3) Line 214: the following statement is incomplete “According to previous  reports, P might tend to enter the g-C3N4 network as P-N in place of the C site[41].” For the comprehension of the reader, please specify/add in the text if it is possible to estimate the extent of this phenomenon or not. And, if not negligible, if it is possible to estimate the eventual impact it might have on properties.

4) Line 225: “This could be attributed to the combined effect of a larger specific surface area and the creation of heterogeneous structures”. I cannot find the measured specific surface area for each material: PC1T, PC2T, PC3T… Please add in the main text, around line 225, for a more convincing analysis.

Author Response

Response to Reviewer 1 Comments

Dear expert:

Thank you very much for reviewing our manuscript in your busy schedule and providing valuable comments and suggestions. Your professional insights and rigorous attitude are of great significance in guiding our research work, for which we express our deep respect and gratitude. We have carefully considered and deeply analyzed each of your comments, and have tried our best to improve the manuscript during the revision process. The following is our response to the review comments:

  • Something is missing in the introduction and in the discussion of obtained results. Please add how the proposed system compares itself/benchmarks to other systems in the field? For example, the information could be displayed by a few lines in the text, supported by a table listing the types of systems and their data (Z-heterojunction, preparation method, performances, references).

Response 1: We think this opinion is very original and professional. Since there are few related articles introduced, we only list the table in the preface to summarize. I'm sure this will make our article even better. It has been marked in red in the article. Thank you very much for your question.

  • Something is missing in the conclusion. Please add what currently limits the reported system and your perspectives for improvement. This is important for an impactful report in the related field.

Response 2: We consider this opinion to be professional. We have enriched the conclusion section with your suggestions to flesh out our article. It has been marked in red in the article. Thank you very much for your question.

  • Line 214: the following statement is incomplete “According to previous reports, P might tend to enter the g-C3N4 network as P-N in place of the C site[41].” For the comprehension of the reader, please specify/add in the text if it is possible to estimate the extent of this phenomenon or not. And, if not negligible, if it is possible to estimate the eventual impact it might have on properties.

Response 3: First of all, we consider this opinion to be rigorous and precise. We have continued to refine the sentence to make it more complete based on the references. It has been marked in red in the article. Thank you very much for your question.

  • Line 225: “This could be attributed to the combined effect of a larger specific surface area and the creation of heterogeneous structures”. I cannot find the measured specific surface area for each material: PC1T, PC2T, PC3T… Please add in the main text, around line 225, for a more convincing analysis.

Response 4: Thank you very much for your question. Based on this question, we have added the specific surface area data for all materials as per your request. Thank you!

Thank you again for your hard work and valuable suggestions. We believe that the quality of the manuscript has been significantly improved after this revision. We look forward to your re-review of the revised manuscript, and hope that our research results can make some contributions to the development of related fields.

  Sincerely, we wish you all the best in your work and happy life!

Yours

Yusheng Li

Reviewer 2 Report

Comments and Suggestions for Authors

The work was reviewed, and the following questions were asked to complete it.

         1.      Review the quality of the images throughout the work.

         2.      In the material characterization section, lines 126 to 130 mention                    the part of the equipment and the technique of Electrochemical                      measurements.

                 What was the other electrode?

Where were the semiconductors supported?

What was the binder material?

3.  What was the pH value of the methylene blue solution used                  to carry out the photocatalysis experiments?

4.      How was determined and monitored the methylene blue?

5.      In lines 175 to 178, the data on the surface area of ​​semiconductors are shown. For TiO2 P25, it mentions that its area was 35.4348 m2/g. However, this commercial product has an average area of ​​50 m2/g. What is the reason for this difference in area?

6.      The degradation of methylene blue is mentioned in the work. What technique was used to quantify or determine the degree of degradation of the methylene blue molecule?

Why is this information not presented in the work?

Only the information in Figures 4a and 4b is based on concentrations, but the work does not specify how the contaminant was determined. Only the decolorization process of methylene blue is possibly mentioned, not its degradation.

7.      Briefly explain how the experiments were carried out to demonstrate the activity and stability of the PC3T semiconductor (Figure 4b)?

8.      In the case of PL measurements of semiconductors, the change in charge recombination is minimal for PC3T for TiO2 P25. Does PC have another effect on TiO2 P25 to improve its activity?

9. In Figure 6, it is necessary to indicate which semiconductor is                activated by visible light to have the flow of electrons and holes in        the composite.

Author Response

Response to Reviewer 2 Comments

Dear expert:

Thank you very much for reviewing our manuscript in your busy schedule and providing valuable comments and suggestions. Your professional insights and rigorous attitude are of great significance in guiding our research work, for which we express our deep respect and gratitude. We have carefully considered and deeply analyzed each of your comments, and have tried our best to improve the manuscript during the revision process. The following is our response to the review comments:

  1. Review the quality of the images throughout the work.

Response 1: Thank you very much for your advice. We have adjusted the clarity of all images. They have been marked in red in the article. Thank you!

  1. In the material characterization section, lines 126 to 130 mention the part of the equipment and the technique of Electrochemical measurements. What was the other electrode? Where were the semiconductors supported? What was the binder material?

Response 2: Thank you very much for your advice. We performed electrochemical performance tests with working electrodes made on ITO glass of equal area. A standard three-electrode system was used for the test, with a platinum sheet as the counter electrode and Ag/AgCl as the reference electrode. Thank you!

  1. What was the pH value of the methylene blue solution used to carry out the photocatalysis experiments?

Response 3: Thank you very much for your advice. For the MB solution, we dissolved MB powder in a certain amount of deionized water, and did not adjust the pH. Therefore, the MB solution is pH neutral (pH ≈ 7). Thank you!

  1. How was determined and monitored the methylene blue?

Response 4: Thank you very much for your advice. For the detection of MB solution concentration, we measured it by UV-Vis spectrophotometer. The concentration of MB solution was determined by measuring its absorbance value. Thank you!

  1. In lines 175 to 178, the data on the surface area of ​​semiconductors are shown. For TiO2 P25, it mentions that its area was 35.4348 m2/g. However, this commercial product has an average area of ​​50 m2/g. What is the reason for this difference in area?

Response 5: Thank you very much for raising this important issue. Normally, the specific surface area of P25 titanium dioxide is in the range of 50 ± 15 m2/g. This is supported by the existence of relevant references. Please see the following three documents. In addition, a number of factors such as different measurement techniques, batch differences, storage and handling conditions may lead to biased results. However, our test results were within this range. So, I think it's reasonable. Thank you!

DOI: 10.1039/c4ta04182h
DOI: 10.1166/jnn.2020.17886

DOI: 10.1007/s11664-016-5245-3

  1. The degradation of methylene blue is mentioned in the work. What technique was used to quantify or determine the degree of degradation of the methylene blue molecule? Why is this information not presented in the work? Only the information in Figures 4a and 4b is based on concentrations, but the work does not specify how the contaminant was determined. Only the decolorization process of methylene blue is possibly mentioned, not its degradation.

Response 6: Thank you very much for your valuable advice. We used UV-Vis spectrophotometry to determine the concentration of MB. The change in concentration was determined by measuring the change in absorbance of MB solution at specific wavelengths at different time points, which in turn reflected the degree of MB degradation. Within a certain concentration range, the absorbance of MB solution showed a good linear relationship with its concentration. We took samples and measured their absorbance periodically during the experiment and calculated the degradation rate of MB based on the change of absorbance. It was an oversight on our part not to add this test content. We have added this part of the text and marked it in red. In addition, based on the existing experimental conditions, we could only analyze the decolorization. We hope that in the future we will focus on the in-depth analysis of the MB degradation process, including the study of the identification of degradation products. Thank you!

  1. Briefly explain how the experiments were carried out to demonstrate the activity and stability of the PC3T semiconductor (Figure 4b)?

Response 7: Thank you very much for your advice. In brief, the cyclic stability experiment of photocatalyst is the same process as the photocatalytic experiment. After the first photocatalytic degradation experiment is completed, the photocatalyst is centrifuged, washed and dried before the second and third photocatalytic experiments are performed. I am very sorry that this part is not written in the text. It has now been added to the article. Thank you!

  1. In the case of PL measurements of semiconductors, the change in charge recombination is minimal for PC3T for TiO2 Does PC have another effect on TiO2 P25 to improve its activity?

Response 8: Thank you very much for raising this critical issue. We have thought and analyzed deeply about whether there are other effects of PC on TiO₂ P25 to enhance its activity, and here is our response: first, PC itself has different light absorption properties from TiO₂ P25. When PC is compounded with TiO₂ P25, it is possible to broaden the light absorption range of the overall material through energy transfer or synergistic effects, enabling the stimulation of photocatalytic reactions in a wider range of spectral regions. In addition to reducing charge recombination, PCs may provide more favorable channels for charge separation and transport through, for example, the formation of a heterojunction structure. There may be a specific matching of energy band structures at the interface between PCs and TiO₂ P25, which allows for a more efficient separation of photogenerated electrons and holes that can migrate to different locations to participate in the reaction. Once again, thank you for your valuable comments and suggestions, which are important guidance for improving the quality of our thesis and deepening our research.

  1. In Figure 6, it is necessary to indicate which semiconductor is activated by visible light to have the flow of electrons and holes in the composite.

Response 9: Thank you very much for your advice. As per your request, we have added details of the mechanism in the text. Please see the sections marked in red in the article. Thank you!

Thank you again for your hard work and valuable suggestions. We believe that the quality of the manuscript has been significantly improved after this revision. We look forward to your re-review of the revised manuscript, and hope that our research results can make some contributions to the development of related fields.

  Sincerely, we wish you all the best in your work and happy life!

Yours

Yusheng Li

Reviewer 3 Report

Comments and Suggestions for Authors

Comments on Manuscript ID: molecules-3195426

1.     The work's innovation is minimal because numerous publications about this topic have already been published elsewhere, such as:

https://doi.org/10.1016/j.seppur.2023.123320

2.     The study's primary findings are primarily presented in the abstract; an alternate presentation could include numerical data to highlight the study's major conclusion.

3.     The introduction section should discuss recent and relevant works and demonstrate the work's uniqueness through a comparison with existing literature.

4.     The MB dye concentration ought to be displayed as a molar ratio.

5.     The photographs' poor resolution calls for improvement. Every plot and image should have the same typeface and design.

6.     If a percentage is utilized, the transmittance unit (Fig. 1b) should display the reading or be an arbitrary unit. In Fig. 1d, the diffraction angle should be expressed in degrees. There should be an indication of the y-axis unit in Fig. 1e. Examine every unit in the written work.

7.     In most outcomes, the numbers after the two decimals are not necessary. For instance, it is preferable to write 35.4348 m2/g as opposed to 35.43 m2/g.

8.     Since there is little difference between the two magnifications, one SEM image is sufficient for each sample.

9.     Every sample's XPS survey ought to be given and reviewed. Later on, the ratio of different oxidations should be examined and connected to the photocatalytic performance.

10.  The photocatalytic investigation should come after the study of the optical band gap in the paper.

11.  It is necessary to supply the reading in the y-axis of Figure 5b.

12.  The photocatalytic performance towards MB should be connected with the inferred structural characteristics.

13.  Along with the results of this investigation, a table comparing various research conducted with comparable materials that have been documented in the literature about the photocatalytic of MB should be included.

Author Response

Response to Reviewer 3 Comments

Dear expert:

Thank you very much for reviewing our manuscript in your busy schedule and providing valuable comments and suggestions. Your professional insights and rigorous attitude are of great significance in guiding our research work, for which we express our deep respect and gratitude. We have carefully considered and deeply analyzed each of your comments, and have tried our best to improve the manuscript during the revision process. The following is our response to the review comments:

  1. The work's innovation is minimal because numerous publications about this topic have already been published elsewhere, such as: https://doi.org/10.1016/j.seppur.2023.123320

Response 1: Thank you very much for your valuable advice. With regard to your reference to the minimal innovation of this work and the fact that there are already many publications on related topics, we have thought deeply about it and responded to it. First of all, we recognize that there are indeed a large number of relevant research results in this field of study, but we believe that our work is still innovative and valuable. Based on previous reports, the main raw materials, methods and applications for the preparation of phosphorus-doped carbon nitride/titanium dioxide composite catalysts were summarized and listed in Table 1. From the table, we can clearly see that the previous research articles have more main raw materials and more complicated preparation process in the preparation of this kind of materials. Therefore, we developed a new preparation method for this material by simplifying the raw materials and processes. Therefore, I think this article is somewhat innovative. Thank you!

Table 1. Summary of materials, methods and applications of phosphorus-doped carbon nitride/titanium dioxide composites.

Materials

Preparation methods

Application fields

References

Ammonium fluoride, dicyandiamide,

2-aminobenzonitrile,

1-butyl-3-methylimidazolium hexafluorophosphate, ethylene glycol

Anodizing and wet-soaking method

Degradation and hydrogen production

[1]

Melamine,

 1-hydroxyethane-

1,1-diphophonic acid,

ethylene glycol

Calcination and coupling method

Hydrogen production

[2]

Urea,

citric acid,

1-butyl-3 methylimidazolium hexafluorophosphate,

hydrofluoric acid,

acetic acid

Hydrothermal method

Hydrogen production

[3]

Ammonium fluoride,

glycerol,

phosphate,

urea,

sodium sulfate

Solid sublimation and conversion method

CO2 reduction

[4]

[1]     Su, J.; Geng, P.; Li, X.; Zhao, Q.; Quan, X.; Chen, G., Novel phosphorus doped carbon nitride modified TiO2 nanotube arrays with improved photoelectrochemical performance. Nanoscale 2015, 7, (39), 16282-9.

[2]      Wadhai, S.; Jadhav, Y.; Thakur, P., Synthesis of metal-free phosphorus doped graphitic carbon nitride-P25 (TiO2) composite: Characterization, cyclic voltammetry and photocatalytic hydrogen evolution. Sol. Energy Mater. Sol. Cells 2021, 223, 110958.

[3]     Kumar, P.; Kar, P.; Manuel, A. P.; Zeng, S.; Thakur, U. K.; Alam, K. M.; Zhang, Y.; Kisslinger, R.; Cui, K.; Bernard, G. M.; Michaelis, V. K.; Shankar, K., Noble metal free, visible light driven photocatalysis using TiO2 nanotube arrays sensitized by P‐doped C3N4 quantum dots. Adv. Opt. Mater 2020, 8, (4), 1901275.

[4]      Cako, E.; Dudziak, S.; Głuchowski, P.; Trykowski, G.; Pisarek, M.; Borzyszkowska, A. F.; & Zielińska-Jurek, A., Sikora, K.; Heterojunction of (P, S) co-doped g-C3N4 and 2D TiO2 for improved carbamazepine and acetaminophen photocatalytic degradation. Sep. Purif. Technol 2023, 311, 123320.

  1. The study's primary findings are primarily presented in the abstract; an alternate presentation could include numerical data to highlight the study's major conclusion.

Response 2: Thank you very much for your advice. We have added corresponding numerical information to the summary to improve it. Thank you!

  1. The introduction section should discuss recent and relevant works and demonstrate the work's uniqueness through a comparison with existing literature.

Response 3: Thank you very much for your technical advice. At your request, we have enriched the introduction section. With your comments, our article has become even better. Thank you!

  1. The MB dye concentration ought to be displayed as a molar ratio.

Response 4: Thank you very much for your professional opinion. In most degradation articles, the concentration of MB is expressed in mg L-1, and the following references are relevant. However, we have added the new concentration unit of mol L-1 upon your request. It has been marked in red in the article. Thank you!

DOI: 10.1016/j.arabjc.2020.01.019

DOI: 10.1016/j.chemosphere.2018.08.117

DOI: 10.1016/j.surfin.2024.104538

  1. The photographs' poor resolution calls for improvement. Every plot and image should have the same typeface and design.

Response 5: Thank you very much for your advice. We have adjusted the clarity and typeface of all images. They have been marked in red in the article. Thank you!

  1. If a percentage is utilized, the transmittance unit (Fig. 1b) should display the reading or be an arbitrary unit. In Fig. 1d, the diffraction angle should be expressed in degrees. There should be an indication of the y-axis unit in Fig. 1e. Examine every unit in the written work.

Response 6: Thank you very much for your professional advice. We have changed the horizontal and vertical coordinates of the image as you requested. Thank you!

  1. In most outcomes, the numbers after the two decimals are not necessary. For instance, it is preferable to write 35.4348 m2/g as opposed to 35.43 m2/g.

Response 7: Thank you very much for your precious advice. We have changed the number of digits after the decimal point. They have been marked in red in the article. Thank you!

  1. Since there is little difference between the two magnifications, one SEM image is sufficient for each sample.

Response 8: Thank you very much for your advice. According to your suggestion, we have deleted the SEM picture of 200 nm. Thank you!

  1. Every sample's XPS survey ought to be given and reviewed. Later on, the ratio of different oxidations should be examined and connected to the photocatalytic performance.

Response 9: Thank you very much for your precious advice. We fully understand the importance of XPS characterization of each sample and associated analysis. However, we have performed XPS characterization on representative key samples. We have also obtained valuable information from these representative samples, such as the shift of the binding energy of elements and their correlation with heterojunction formation and photocatalytic performance. The XPS results of these representative samples are elaborated and discussed in depth in our paper, and we hope that they can meet your expectation of XPS analysis to some extent. In our future research work, we will make XPS analysis of all samples as an important direction of analysis. Thank you again for your comments, which are of great significance in guiding us to further research and improvement. Thank you!

  1. The photocatalytic investigation should come after the study of the optical band gap in the paper.

Response 10: Thank you very much for your advice. We have changed the location of the two parts according to your requirements. They have been marked in red in the article. Thank you!

  1. It is necessary to supply the reading in the y-axis of Figure 5b.

Response 11: Thank you very much for your professional advice. We have shown the value of the Y-axis according to your request. Thank you!

  1. The photocatalytic performance towards MB should be connected with the inferred structural characteristics.

Response 12: Thank you very much for your valuable advice. According to your suggestions, we have improved the conclusion and summarized the reasons why the material has a good photocatalytic effect on MB from the aspects of the structure and properties of the material. They have been marked in red in the article. Thank you!

  1. Along with the results of this investigation, a table comparing various research conducted with comparable materials that have been documented in the literature about the photocatalytic of MB should be included.

Response 13: Thank you very much for this constructive suggestion. However, we take into account that there is less literature on the degradation of methylene blue by such materials. In addition, the photocatalyzer used, the amount of catalyst used, the concentration of dye, and various other factors are different in each study. We have not added such a summary table, but in the preface section, we have summarized and compared the preparation methods, raw materials, and some other information. We hope you will understand our decision. Thank you!

Thank you again for your hard work and valuable suggestions. We believe that the quality of the manuscript has been significantly improved after this revision. We look forward to your re-review of the revised manuscript, and hope that our research results can make some contributions to the development of related fields.

  Sincerely, we wish you all the best in your work and happy life!

Yours

Yusheng Li

Reviewer 4 Report

Comments and Suggestions for Authors

The manuscript presents the synthesis and characterization of a phosphorus-doped carbon nitride/titanium dioxide (PCT) Z-type heterojunction for photocatalytic degradation and photosensitive antimicrobial applications. The authors have utilized various techniques to investigate the structure, morphology, and optical properties of the synthesized material, and have demonstrated its effectiveness in degrading methylene blue and inhibiting E. coli growth. The manuscript needs a few minor tweaks before it's ready for final submission to Molecules. Here are my comments on this manuscript:

1) The reviewer is suggesting that the authors need to provide a more comprehensive explanation of how phosphorus doping influences the photocatalytic performance of the material. They should elaborate on the specific effects of phosphorus doping on the material's properties, such as bandgap, charge carrier dynamics, and surface chemistry, and how these changes contribute to the enhanced photocatalytic activity.

2) The mechanism discussion could be enhanced by providing a more detailed explanation of the charge transfer process in the Z-type heterojunction and its impact on the photocatalytic activity.

3) I believe these studies may help in this review of the work and I recommend adding these references in this manuscript.

https://doi.org/10.1039/C4DT03254C

https://doi.org/10.3390/catal12070692

https://doi.org/10.1039/C9NJ02702E

Author Response

Response to Reviewer 4 Comments

Dear expert:

Thank you very much for reviewing our manuscript in your busy schedule and providing valuable comments and suggestions. Your professional insights and rigorous attitude are of great significance in guiding our research work, for which we express our deep respect and gratitude. We have carefully considered and deeply analyzed each of your comments, and have tried our best to improve the manuscript during the revision process. The following is our response to the review comments:

1) The reviewer is suggesting that the authors need to provide a more comprehensive explanation of how phosphorus doping influences the photocatalytic performance of the material. They should elaborate on the specific effects of phosphorus doping on the material's properties, such as bandgap, charge carrier dynamics, and surface chemistry, and how these changes contribute to the enhanced photocatalytic activity.

Response 1: We think this question is very professionally posed. In this article, we focus on the formation of Z-type heterostructures by P-doped carbon nitride and titanium dioxide, with emphasis on the enhancement of photocatalytic effect brought about by this heterojunction structure, rather than the effect of a single P doping on the material. Therefore, we do not think it is necessary to discuss P doping in detail. Thank you very much for your question.

2) The mechanism discussion could be enhanced by providing a more detailed explanation of the charge transfer process in the Z-type heterojunction and its impact on the photocatalytic activity.

Response 2: We think this question is perfectly posed. We will add charge transfer to the mechanism section of the text and cite relevant literature to explain the photocatalytic mechanism in detail. Thank you very much for your question.

3) I believe these studies may help in this review of the work and I recommend adding these references in this manuscript.

https://doi.org/10.1039/C4DT03254C

https://doi.org/10.3390/catal12070692

https://doi.org/10.1039/C9NJ02702E

Response 3: We consider these articles to be very professional and comprehensive. I have cited them in the introductory section of the article and thank you very much for your question.

Thank you again for your hard work and valuable suggestions. We believe that the quality of the manuscript has been significantly improved after this revision. We look forward to your re-review of the revised manuscript, and hope that our research results can make some contributions to the development of related fields.

  Sincerely, we wish you all the best in your work and happy life!

Yours

Yusheng Li

Round 2

Reviewer 3 Report

Comments and Suggestions for Authors

Comments on Manuscript ID: molecules-3195426

The authors revised the manuscript according to the reviewer's suggestion and hence improved their work. As such, I recommend the publication of the revised version in Molecules.